*Correspondence*

# Response to Suresh et al

Anumeha Singh[1,2], Lekha E Manjunath[1,2], Md Noor Akhtar [1,2], Japita Ghosh [1,2] &
Sandeep M Eswarappa [1✉]

Reply to: AN Suresh et al
See also: LE Manjunath et al and A Singh et al

In this correspondence, we would like to argue, with experimental evidence, that the conclusions and interpretations of Suresh et al (2025) are incorrect. As we explain below, their experiments lack rigor, critical controls, and therefore, the results are not sufficiently conclusive. More importantly, the authors have overlooked multiple lines of both direct and indirect published evidence—provided by our group as well as others—that demonstrate physiologically relevant levels of stop codon readthrough (SCR) of *AGO1* (Bidou et al, 2022; Del Toro et al, 2021; Ghosh et al, 2020; Singh et al, 2019). Surprisingly, Suresh et al do not cite these publications except the ones from our laboratory.

## Multiple lines of evidence for SCR of *AGO1*

Suresh et al have missed one of our key results. Contrary to their claim, we have performed the standard single-mRNA dual-luciferase assay, which was used by multiple groups including Loughran et al in their previous studies (Grentzmann et al, 1998; Loughran et al, 2014). And using this, we have demonstrated the SCR of *AGO1* with an efficiency of ~7% (in cells) and ~9% (in vitro cell-free system), which was significantly above the background level shown in a negative control (which is lacking in the results shown by Suresh et al) (Fig 4B in Singh et al (2019)).

These observations are consistent with *AGO1* SCR demonstrated by fluorescence reporter-based assay and Myc-tag based western blotting assays in the same study (Singh et al, 2019). Notably, the endogenous SCR product (Ago1x) constitutes ~40% of total Ago1 isoforms, and it was identified in multiple mouse tissues. Analysis of publicly available mass-spectrometry data revealed peptides that can be generated only by SCR of *AGO1* in tissue samples from mouse, which is a direct evidence of SCR of endogenous *AGO1*. Ribosome profiling data analysis also revealed significant ribosome footprints (in-frame with the coding sequence footprints) in the proximal 3′ UTR of *AGO1* in multiple cell types, supporting >10% SCR efficiency.

In another independent study, Ghosh et al have provided direct evidence for SCR of endogenous *AGO1* by detecting the peptides specific to the readthrough product of *AGO1* (Ghosh et al, 2020), and by detecting the SCR product using a specific antibody (raised independently in their laboratory). The study also demonstrates the physiological significance of this process by deleting the genomic region that encodes the proximal 3′UTR responsible for SCR. These mutant cell lines that lack SCR of *AGO1* showed reduced proliferation (Ghosh et al, 2020). These findings, which provide a strong, independent evidence for physiologically significant SCR of endogenous *AGO1*, have been overlooked by Suresh et al.

Using ribosome profiling method, Bidou et al have reported signal of *AGO1* SCR (along with *VEGFA*, *MTCH2*, *LDHB*, and *MDH1*) in the form of significant ribosome footprints after the stop codon even in the absence of any readthrough inducing agent (Bidou et al, 2022). This study too has not been cited by Suresh et al.

## Experiments shown in Suresh et al are not sufficiently conclusive

The assays shown in Suresh et al lack important negative controls, i.e., a construct which does not show readthrough. This control is essential to know the background luminescence and basal readthrough levels, and to conclude if the test construct shows readthrough activity more than the basal levels. This is even more important to conclude lack of SCR in any mRNA, when multiple independent lines of evidence are in favor of it. In our studies, we have used constructs that lack the readthrough element or constructs with a random sequence as negative controls. *AGO1* shows SCR efficiency significantly higher than these negative controls (Singh et al, 2019).

Reporter systems suffer from many potential biases such as alternative promoter, alternative splicing, leaky scanning, reinitiation, etc. They must be rigorously investigated before coming to a conclusion on readthrough or lack of it. In our study, we have included negative controls, checked mRNA levels, performed western blotting and in vitro SCR assays to rule out the potential artefacts mentioned above (Manjunath et al, 2024). The assays performed by Suresh et al have not investigated any of these possibilities, especially for their positive control to demonstrate that the signal they are detecting is actually due to SCR.

This is particularly important as stop-go (SG) sequence can itself affect the translation of the downstream region (F-Luc in this case) because of ribosomal pausing at the end of the SG sequence (Donnelly et al, 2001; Doronina et al, 2008). In fact, a significant number of ribosomes (up to 60%) could fall off from the site resulting in discontinuation of translation (Liu et al, 2017). Thus, inserting SG sequence after R-Luc (Fig. 1A in Suresh et al, 2025) can negatively influence SCR.

We tested this hypothesis using *HBB* (β-globin) mRNA, which has been shown to exhibit SCR in multiple studies (Chittum et al, 1998; Geller and Rich, 1980; Hatfield et al, 1988). The regular dual-luciferase assay

[1]Department of Biochemistry, Indian Institute of Science, Bengaluru, India. [2]These authors contributed equally: Anumeha Singh, Lekha E Manjunath, Md Noor Akhtar, Japita Ghosh. ✉E-mail: sandeep@iisc.ac.in

https://doi.org/10.1038/s44318-025-00479-0 | Published online: 11 June 2025

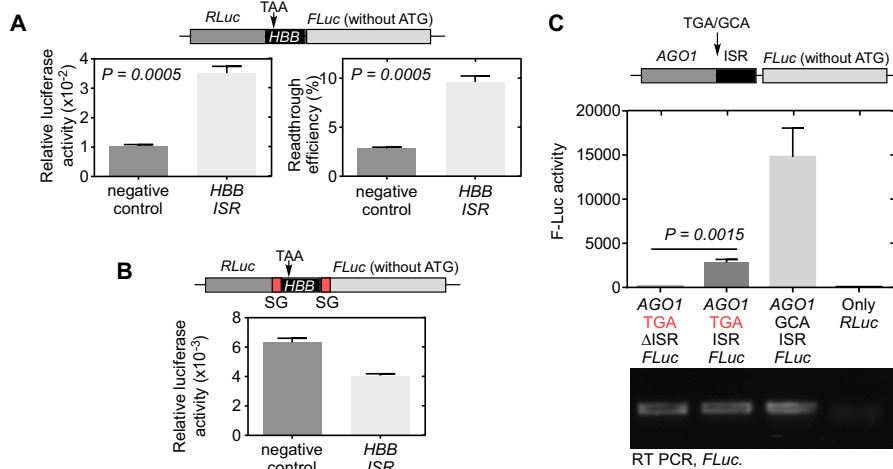

**Figure 1. Limitation of luciferase reporters with SG sequence for SCR assay.**

(A, B) SCR assay was performed in HEK293 cells using dual luciferase constructs without (A) or with (B) SG sequences as indicated in the schematics. 30 nucleotides of the distal coding sequence of human *HBB*, its stop codon (TAA), and 60 nucleotides of its proximal 3′UTR (ISR) were cloned between and in frame with R-Luc and F-Luc. Relative luciferase activity, indicative of SCR, is plotted in the graphs. In (A), readthrough efficiency expressed as % of the activities obtained from an in-frame control without any stop codon between R-Luc and F-Luc is also shown. Negative controls had the sequence 5′ TCCAGTGTGGTGGAATTCTGCAGATATCCA 3′ in the same place as *HBB* sequence and it was cloned in frame with R-Luc and F-Luc. (C) F-Luc activity observed in HEK293 cells transfected with *AGO1*-FLuc constructs as indicated. The data are taken from the Fig. 1C of Singh et al, 2019. All bars indicate mean ± SE, N = 3. P value, two-sided student's t-test. ISR, inter stop codon region in the proximal 3′UTR responsible for SCR. Detailed methods is provided in Appendix and luciferase readings are provided in source data file. Source data are available online for this figure.

without SG sequences in the construct demonstrated SCR of *HBB* significantly above the background levels (~10% efficiency). However, the pSGDlucV3.0 dual-luciferase assay system with SG sequences used by Suresh et al failed to demonstrate the same in HEK293 cells (Fig. 1A,B). These observations show that luciferase reporters with SG sequence are not reliable to test SCR.

In a previous study, this SG-sequence-based dual-luciferase assay failed to detect the SCR of *VEGFA* (Loughran et al, 2017). However, there are strong evidence for SCR of *VEGFA* from multiple studies from different research groups (Bidou et al, 2022; Eswarappa et al, 2014; Liu et al, 2023; Manjunath et al, 2024; Nico et al, 2021; Singh et al, 2019). Therefore, conclusions based just on this assay can be misleading.

Another reason for the apparent lack of SCR in their assay (Suresh et al, 2025; Fig. 1) is the short readthrough element they have used. In all our single-mRNA dual-luciferase SCR assays, we have used 99 nucleotides of the proximal 3′UTR of *AGO1* as test element to drive SCR. However, Suresh et al have used only a part of the readthrough element.

Overall, incomplete readthrough element and presence of SG sequence that can cause ribosomal fall off are likely factors responsible for the apparent lack of *AGO1* SCR in their assays. Furthermore, lack of negative controls makes their experiments inconclusive.

## Misinterpretation of our results by Suresh et al

Suresh et al suggest that our reporters have intrinsic F-Luc activity. The data shown in Fig. 1C (taken from Fig. 1C of Singh et al, 2019) clearly provides evidence against their interpretation. Only in the presence of the readthrough element, there is F-Luc activity above the background level. If there was an intrinsic F-Luc activity, it would have been there in both the constructs (with or without ISR, first two bars in Fig. 1C).

Suresh et al incorrectly mention that 20% SCR efficiency is the highest reported efficiency. However, their group has previously demonstrated SCR of *OPRL1* at 31% efficiency (Loughran et al, 2014). The corresponding value for *AGO1*, obtained by similar assay is ~7% (Singh et al, 2019).

Their interpretation that F-Luc activity reduces when fused to AGO1 is incorrect. In

our single-mRNA dual-luciferase assay the F-Luc is not fused with AGO1, yet it shows efficient SCR significantly above the basal levels shown by a negative control (Singh et al, 2019). Furthermore, Suresh et al have wrongly interpreted that our conclusions are based on calculation of readthrough relative to a compromised sense control (fused). As presented in Singh et al (2019), our conclusions on SCR are primarily based on the reporter signal above the background levels shown by negative controls.

In summary, the experiments performed by Suresh et al do not refute SCR of *AGO1*, which is based on multiple lines of evidence reported from different laboratories. Therefore, by not detecting SCR of *AGO1*, Suresh et al's study has actually demonstrated the limitations of the SG sequence-based luciferase assay in investigating SCR. We propose that the results of this reporter assay should be verified by other methods to rule out or demonstrate SCR. Techniques such as western blotting, mass-spectrometry and ribosome profiling coupled with advanced computational tools are more reliable as they detect SCR of endogenous mRNAs, which is physiologically relevant. As described above, *AGO1* has passed all these tests in multiple laboratories.

## Methods

### Construction of plasmids for SCR assays

The single-mRNA dual-luciferase assay construct without stop-go sequence is described previously (Singh et al, 2019). Thirty nucleotides of the distal coding sequence of human *HBB*, its stop codon, and 60 nucleotides of its proximal 3′UTR were cloned between and in frame with Renilla luciferase (R-Luc) and Firefly luciferase (F-Luc) using enzyme sites *Bam*HI and *Xho*I. To calculate the SCR efficiency, an in-frame control was constructed by mutating the stop codon (TAA) of *HBB* to GCA.

For single-mRNA dual-luciferase assay construct with stop-go sequence, we procured pSGDlucV3.0 from Addgene (#119760). This is the same plasmid used by Suresh et al. In this plasmid, thirty nucleotides of the distal coding sequence of human *HBB*, its stop codon, and 60 nucleotides of its proximal 3′UTR were cloned between and in frame with R-Luc and F-Luc using enzyme sites *Hind*III and *Bgl*II. In all *HBB* constructs, the stop codon of R-Luc was removed as the canonical stop codon of *HBB* was present. Negative controls

had the sequence 5′-TCCAGTGTGGTGGA ATTCTGC AGATATCCA-3′ in the same place as *HBB* sequence and it was cloned in frame with R-Luc and F-Luc.

## SCR assay

The plasmid constructs (0.5 µg per well of a 24-well plate) were transfected in HEK293 cells using polyethylenimine (PEI, 2 µg/mL). Cells were harvested after 24 h of transfection and R-Luc and F-Luc activities were measured in Glomax Explorer (Promega) using Dual-Luciferase Reporter Assay System (Promega). Relative luciferase activity was calculated by taking the ratio of F-Luc and R-Luc. The SCR efficiency was calculated by the following formula:

$$\frac{\left(\frac{RLuc}{FLuc}\right)test}{\left(\frac{RLuc}{FLuc}\right)no-stop} * 100$$

## List of primers

For cloning *HBB* sequence in pSGDlucV3.0:
5′-ATGCAAGCTTAGTGGCTAATGCC CTGGCCCA-3′ and
5′-ATGCAGATCTGTAGTTGGACTTA GGGAACA-3′
For cloning *HBB* sequence in pCDNA3.1 dual luciferase construct:
5′-ATGCGGATCCGTGGCTAATGCC-3′
5′-ATGCCTCGAGGTAGTTGGACTTA GG-3′

## Peer review information

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

## Author contributions

**Anumeha Singh**: Conceptualization; Formal analysis; Investigation. **Lekha E Manjunath**: Conceptualization; Formal analysis. **Md Noor Akhtar**: Formal analysis; Investigation. **Japita Ghosh**: Formal analysis; Investigation. **Sandeep M Eswarappa**: Conceptualization; Resources; Data curation; Formal analysis; Writing—original draft; Project administration.

## Disclosure and competing interests statement

The authors declare no competing interests.

