## [Peer Review File · The EMBO Journal]

Response to Suresh et al.

Anumeha Singh, Lekha Manjunath, Md Noor Akhtar, Japita Ghosh, and Sandeep Eswarappa

Corresponding author(s): Sandeep Eswarappa (sandeep@iisc.ac.in)

Review Timeline:

Submission Date:	24th Mar 25
Editorial Decision:	10th Apr 25
Revision Received:	18th Apr 25
Accepted:	13th May 25

Editor: Cornelius Schneider

Transaction Report:

Dear Sandeep,

Thank you very much for your message. Yes, you can include the additional data. However, please make sure that rebuttal is short and to the point as it is already significantly longer than usual correspondences (Please aim to stay within 4 pages with your current text size settings and with a maximum of 1 figure). We think that your arguments are well reasoned, but we also feel that some of the paragraphs are repetitive, and certain statements use excessively strong language which we think would not benefit your arguments (E.g., instead of saying "...are non-conclusive", saying "are not sufficiently/fully conclusive", etc. Instead of "is being ignored" say, "appears to have been overlooked/missed" etc).

In addition, we have several more formal editorial requirements before we can export the manuscript:

- Please remove the figures from manuscript file and upload as an individual Figure file, figure legend should be placed below the References
- Please place the author list should be placed below the title, affiliations are missing
- Please add a "DISCLOSURE AND COMPETING INTERESTS STATEMENT"
- Please correct the section order: Title page - Acknowledgements (if any) - Disclosure and Competing Interests Statement - References - Figure Legend - Table (if any).
- Please provide source data for the newly added experiments (<https://www.embopress.org/sourcedata>). Source data files need to be saved in a scheme one figure/folder and then uploaded as .zip files. The Source data files for figure 1 need to be saved in a single folder and this needs to be zipped and then uploaded as "SD figure 1.zip" file.

Would it be possible to re-submit the revised manuscript as soon as possible, latest early next week?

With best regards,

Cornelius

Cornelius Schneider, PhD
Editor | The EMBO Journal
c.schneider@embojournal.org

Use the link below to submit your revision:

Response to Reviewers' comments

Referee #1:

However, in Singh et al., 2019, the sensor assays were the most convincing data for me and the only data indicating a miRNA-dependent regulation.

Our Response: We respectfully disagree that sensor assays are the only data indicating miRNA-dependent regulation. The Reviewer has missed Fig. 3B and 3C in Singh et al 2019, which show miRNA-dependent regulation of endogenous AGO1. There we show evidence for regulation of readthrough using miRNA mimics and inhibitors.

In summary, I believe that publishing this correspondence will be of great benefit to a broad audience and further discussions. In future studies of readthrough at AGO-1, all parties might want to consider changing the tissue culture cell line from HEK-293 to MDA-MB-231 (Ghosh et al., 2020). With the low activity of readthrough at AGO-1 in HEK-293 cells, the attentive reader will have an opportunity to re-evaluate regulatory data presented by the Eswarappa group in 2019 and 2024. The Eswarappa group might want to confirm these observations in more robust cell systems.

Our Response: The Reviewer appears to have missed multiple experiments with HeLa cells in our Singh et al 2019 paper. Many of the experiments on characterization of AGO1 SCR and its regulation by Let-7a were performed in HeLa cells as they show robust expression of Let-7a (Fig 3, Singh et al). Ghosh et al also have detected endogenous Ago1x in HeLa cells (Fig 1D, Ghosh et al) using a different antibody. However, they have not performed readthrough reporter assays in any cells. Their inability to detect endogenous Ago1x in HEK293 cells cannot be concluded as evidence of absence of SCR as the assay depends on the affinity of the antibody they have raised independently.

Referee #2:

Comment: Unfortunately, I was unable to find the table EV1 mentioned.

Response: This is the Table EV1 from Singh et al 2019:

Table EV1. Details of ribosome profiling datasets which showed positive evidence for translational readthrough of AGO1

Cell type	N	RPBM Values (Mean \pm S.E.) (10^{-3})			Ratio of ISR to CDS	Ratio of ISR to 3'UTR
		CDS	ISR	3'UTR		

U2OS	16	3.89 ± 0.86	0.41 ± 0.08	0.02 ± 0.006	0.11	20.5
B cells	2	1.49 ± 0.24	1.03 ± 0.35	0.14 ± 0.03	0.69	7.36
HCT116	2	1.21 ± 0.13	0.36 ± 0.26	0.03 ± 0.002	0.30	12
Huh-7	2	5.85 ± 2.21	5.06 ± 4.49	1.21 ± 1.17	0.86	4.18
MCF7	2	1.48 ± 0.02	0.20 ± 0.04	0.01 ± 0.002	0.14	20
HeLa	1	1.83	0.03	0.002	0.02	15
HEK293	1	1.62	0.40	0.03	0.25	13.333
MEF	1	1.56	0.34	0.008	0.22	42.5

2- The argument regarding the lack of negative controls is questionable. While Loughran et al. included in-frame controls (positive controls), the necessity of negative controls in this specific scenario requires further consideration. Since they were already unable to detect readthrough activity from the Ago1 sequence, it's unclear what information a negative control would provide.

Response: We would like to respectfully disagree. This control is essential to know the background luminescence and basal readthrough levels, and to conclude if the test construct shows readthrough activity more than the basal levels. This is even more important to conclude lack of SCR in any mRNA, when multiple other lines of evidence are in favour of it. In all our assays, we have used constructs that lack the readthrough element or constructs with a random sequence as negative controls. AGO1 shows SCR efficiency significantly higher than these negative controls (Fig 1C, 1F, 4B in Singh et al).

Also, it should be noted that the data shown by Suresh et al in Fig 1 shows % of readthrough, not the absolute luciferase activities nor the relative luciferase activities (FLuc/RLuc). In such calculations, percentages can be misleading as they can vary based on whether the luciferase readings are within the linear range of detection or not. For example, one can get erroneously high % SCR when the luciferase readings are at background levels. Therefore, it is important to have a negative control, which will define the background readthrough activity and forms the basis for testing the gene of our interest.

3- Although it is true that RiboSeq data from Bidou et al, detects readthrough in Ago1 this is only clear in presence of G418, a readthrough inducer, so this is not a strong argument in favor of SCR in Ago1. However the various cell lines used can also account for discrepancy.

Response: We would like to quote from Bidou et al paper, "We detected readthrough signals on DHX38, LDHB, MDH1, MTCH2, VEGFA, and AGO1 genes, with or without G418, which are among the genes known to undergo programmed TC readthrough".

Therefore, evidence of AGO1 was found in HeLa cells even in the absence of readthrough-inducing agent in the above study.

4- I concur that reporter plasmids can be a source of misinterpretation and require careful application. However, the potential pitfalls outlined here would typically lead to overestimation, not underestimation, of activity. Therefore, I am not persuaded by the argument that these limitations automatically explain the lack of observed readthrough.

Response: Yes, it can result in overestimation of luciferase readings. But, this can lead to erroneous conclusions, like absence of SCR, when the positive control and the in frame (sense) control show these aberrant expressions not found in the test constructs. This is particularly important as SG sequence can result in potential biases. The SG sequence can itself affect the translation of the downstream region (F-Luc in this case) because of ribosomal pausing at the end of the SG sequence (Doronina et al, 2008). In fact, a significant number of ribosomes (up to 60%) could fall off from the site (aided by release factors) resulting in discontinuation of translation (Liu et al, 2017). Thus, inserting SG sequence after R-Luc (as shown in Fig 1A in Suresh et al.) can negatively influence SCR. We have demonstrated this using HBB (beta-globin mRNA) in Fig 1A and B.

5- I did not check the issue with the HindIII site, but clearly differences within the 3' UTR region could account for the absence of readthrough activity. This must be verified carefully.

Response: Authors have now provided the sequence of the construct to verify this. However, in the single-mRNA dual luciferase constructs, they still have partial readthrough element (57 nucleotides), instead of 99 nucleotides which we have used. This can also explain apparent lack of SCR.

6- The argument for alternative initiation is not entirely convincing. If this were the case, we would expect to see similar activity in the presence of the stop codon, not just the sense codon. As previously mentioned, investigating mRNA stability and/or initiation rate would be probably more interesting.

Response: Alternative initiation can result in high luciferase activity in a positive control resulting in an erroneous conclusion that the assay to detect readthrough is working.

In our paper (Singh et al 2019) we have measured mRNA levels in readthrough assays.

In conclusion, discerning between the opposing viewpoints presents a significant challenge. While compelling evidence from mass spectrometry and

other research groups supports AGO1 undergoing stop-codon readthrough with a potential physiological role, the SG-F-Luc reporter system employed by Loughran et al. appears to exhibit inconsistencies. Therefore, quantification of readthrough efficiency using this system requires significant caution.

Response: We agree with this assessment. The SG-F-Luc reporter system should not be considered as a gold standard to declare or refute an SCR event.

Referee #3:

Thus, Suresh et al. manuscript suffers from clear deficiencies that cast doubt on the validity of their findings. Singh et al. correctly and significantly pinpoint these deficiencies, and provide further evidence supporting their published observations. Additionally, their observations were independently supported by publications from other labs. Finally, I am mostly concerned about the flow in the 3'UTR construct made by Suresh et al., which can explain most discrepancies and could have been quickly resolved through direct interaction between the two labs.

Response: We agree with Reviewer's point. The experiments conducted by Suresh et al are not rigorous enough to refute the SCR of AGO1 which has been demonstrated by multiple laboratories.

Dear Prof. Eswarappa,

I am pleased to inform you that your manuscript has been accepted for publication in the EMBO Journal.

Yours sincerely,

Cornelius Schneider, PhD
Editor
The EMBO Journal
c.schneider@embojournal.org
